# Contrastive Language-Image Pre-Training with Knowledge Graphs

**Xuran Pan  Tianzhu Ye  Dongchen Han  Shiji Song  Gao Huang**[*]
Department of Automation, BNRist, Tsinghua University, Beijing, China
`{pxr18, ytz20, hdc19}@mails.tsinghua.edu.cn`
`{shijis, gaohuang}@tsinghua.edu.cn`

## Abstract

Recent years have witnessed the fast development of large-scale pre-training frameworks that can extract multi-modal representations in a unified form and achieve promising performances when transferred to downstream tasks. Nevertheless, existing approaches mainly focus on pre-training with simple image-text pairs, while neglecting the semantic connections between concepts from different modalities. In this paper, we propose a knowledge-based pre-training framework, dubbed *Knowledge-CLIP*, which injects semantic information into the widely used CLIP model [38]. Through introducing knowledge-based objectives in the pre-training process and utilizing different types of knowledge graphs as training data, our model can semantically align the representations in vision and language with higher quality, and enhance the reasoning ability across scenarios and modalities. Extensive experiments on various vision-language downstream tasks demonstrate the effectiveness of Knowledge-CLIP compared with the original CLIP and competitive baselines.

## 1   Introduction

Large-scale vision-language pre-training has attracted wide research interests in recent years [9, 26, 38, 72]. Different from training independent models for each specific task, pre-trained models take the analogy of human biological intelligence system, trying to perceive the world from various data modalities and handle comprehensive tasks. Specifically, it aims to provide a unified inference paradigm that simultaneously learns representations for multi-modal data and can easily transfer to a variety of downstream tasks. Benefiting from the accessibility of massive image-text pairs from the web, the vision-language pre-training can leverage a broader source of supervision, and effectively improves the model's generalization power.

Early attempts on vision-language pre-training mainly focus on detecting objects in the images and aligning the corresponding word tokens with object regions [9, 28, 50]. Though effective, the entanglement with the concept of objects, and the additional resources for pre-trained object detectors impose restrictions on real-world applications. One of the pioneer works, CLIP [38], extends the scale of the pre-training dataset to 400 million image-text pairs, and learns representations by directly matching raw text with the corresponding image. Through a contrastive-based training scheme, CLIP learns visual concepts under a large vocabulary which significantly improves the model performances on various downstream tasks. Taking inspiration from CLIP, the following researches further extend the work from several perspectives, including data modality [72], downstream tasks [57], and training data efficiency [19, 44].

---

[*]Corresponding author.

36th Conference on Neural Information Processing Systems (NeurIPS 2022).

**(a) Results on templates with opposite semantic description**

**(b) Results on templates with wrong semantic description**

Figure 1: CLIP fails to accurately capture some fine-grained semantic information. When given opposite semantic descriptions, *e.g.,* adding 'not' in the template or describing an image with wrong color, CLIP tends to give similar distribution as the correct counterpart. Best view in color.

Although showing promising results, the current pre-training frameworks also suffer from limitations. Specifically, the data pairs for pre-training are organized in the simplest manner, where only the descriptions of *matched* and *unmatched* are used to represent the relation between a given image and text pair. This usually leads to a degenerated scenario, where the model tends to rely on the co-occurrence of inputs instead of their semantic meanings. We give a toy example in Fig. 1 by evaluating the zero-shot transfer performance of CLIP on the ImageNet dataset [10] with the templates 'a photo of a {}' and 'not a photo of a {}'. It is shown that the distributions of CLIP outputs under two templates are quite similar, suggesting that the current model fails to understand the semantic meaning of word tokens. As a result, the transferability of the model is restricted, and tends to show worse performances on tasks that require reasoning ability, *e.g.,* visual question answering.

To address the limitation of pre-trained models on semantic perceiving, we resort to the technique of knowledge graph, which has been widely studied in the field of natural language processing [7, 58]. Knowledge graph (KG) is a large-scale semantic network that comprises entities as nodes and semantic relations as edges. Through organizing data in a graph structure, knowledge graphs provide rich information on describing the relations between entities and enable a reasoning process through the whole graph. These advantages over regular-structured data are favorable on various tasks including question-answering [18, 70], relation prediction [29, 43] and knowledge reasoning [6, 59]. In recent years, knowledge graph has also been investigated in the field of computer vision, *e.g.,* scene graph [65], and the integration of both language and image [2]. This bridges the gap between different modalities in the knowledge graph, which inspires us to explore a new knowledge-based pre-training framework, and inject semantic information into simple image-text pairs.

In this paper, we propose a novel vision-language pre-training approach, dubbed *Knowledge-CLIP*, by constructing a knowledge augmented pre-training framework based on the widely used CLIP models. As illustrated in Fig. 2, we follow the structure of CLIP, and use two Transformer-based models as image and text encoders respectively. These two encoders take entities and relations in the knowledge graph as input and extract raw features for both entities and relations. Notably, entities can be in the form of image/text, while the relations are constantly described by language tokens. Then, a multi-modal Transformer encoder is adopted to fuse the entity features conditioned on their relations. In this way, the pre-trained model is pushed to concentrate on understanding semantic relations between visual and word concepts, thereby establishing strong semantic connections between vision and language modalities.

To additionally improve the training efficiency and avoid the massive computation cost in the pre-training procedure, we adopt a simple continuous learning strategy by training our model based

on the pre-trained weights of CLIP. This provides a possibility of efficiently promoting the model performance of CLIP with low training resources.

We train our model on three knowledge graph datasets, namely Visual-Genome [24] (scene graph), ConceptNet [46] (language-based graph), and VisualSem [2] (multi-modal graph), and also adopt part of datasets from CLIP to avoid the model forgetting problem. With the knowledge-enhanced pre-training, Knowledge-CLIP achieves consistent improvements over the original CLIP models on various vision and language downstream tasks.

## 2 Related works

**Large-scale pre-training.** Benefited from the development of Transformer in both vision [35, 63, 36] and language [54] tasks, large-scale pre-training framework has received wide concerns in recent years and shown promising results in the field of computer vision and natural language processing. GPT [39] is one of the pioneer works for language pre-training which optimizes the probability of output based on previous words in the sequence. BERT [11] adopts the masked language modeling technique and predicts the masked tokens conditioned on the unmasked ones.

Similarly, computer vision society also witnesses the development of pre-training models thanks to the emergence of large-scale image datasets. IGPT [5] proposes a generative pre-training technique and shows promising results on classification task. MAE [17] adopts a similar pre-training scheme as BERT and predicts the masked regions of an image with unmasked ones.

Multi-modal pre-training bears differences from the aforementioned frameworks and requires the alignment between various data modalities. Using enormous image-text pairs collected from Internet, vision-language models show significant improvements on various downstream tasks. Among these approaches, various pre-training scheme is adopted, including contrastive learning [1, 27, 31], masked language modeling [47, 51], and masked region modeling [9].

The problem of semantic misunderstanding has also been investigated by previous works. EI-CLIP [33] considers the problem of cross-modal retrieval in the field of E-commerce. Sharing similar insight with our work, the authors notice the model bias towards some specific word tokens in CLIP, and introduce causal inference to align the text encoder with e-commerce domain knowledge. K3M [73] focuses on the modality-missing and modality-noise problem and introduces knowledge modality into E-commerce tasks. DeVLBert [69] studies the spurious correlations between different modalities and adjusts the conditional probability of image tokens and word tokens. Kaleido-BERT [74] focuses on image-text coherence by introducing several novel self-supervised tasks.

Compared to previous approaches, we are the first to incorporate multi-modal knowledge graphs into the pre-training process, and effectively enhance the model perception on semantic relations between visual and language concepts.

**Knowledge Graph.** Knowledge graph is first introduced in the field of natural language processing, and the knowledge graph embedding approaches have been successful on capturing the semantics of symbols (entities and relations) and achieving impressive results on a wide range of real-world applications including text understanding [13, 66], recommendation system [16, 56] and natural language question answering [18, 70]. On the other hand, scene graphs represent a type of graph-structured data in computer vision, where the visual concepts in the image are connected with semantic relations. Scene graphs emphasize the fine-grained semantic features for images and are widely adopted in various downstream tasks, including scene graph generation [65], and Scene Graph Parsing [68]. Besides scene graph, knowledge graph is also adopted in other computer vision tasks, including image classification [22], panoptic segmentation [62], and image captioning [71]. On this basis, multi-modal knowledge graph earns wide concerns in recent years. Considering the natural alignment between different data modalities, multi-modal knowledge graphs have been widely adopted in various graph-based tasks including link prediction [3, 30], entity classification [61], while also showing great potential on out of graph applications like visual question answering [20, 41] and recommendation systems [49, 52].

## 3 Contrastive Language-Image Pre-training (CLIP)

We first provide a brief review of model architectures and training settings in CLIP.

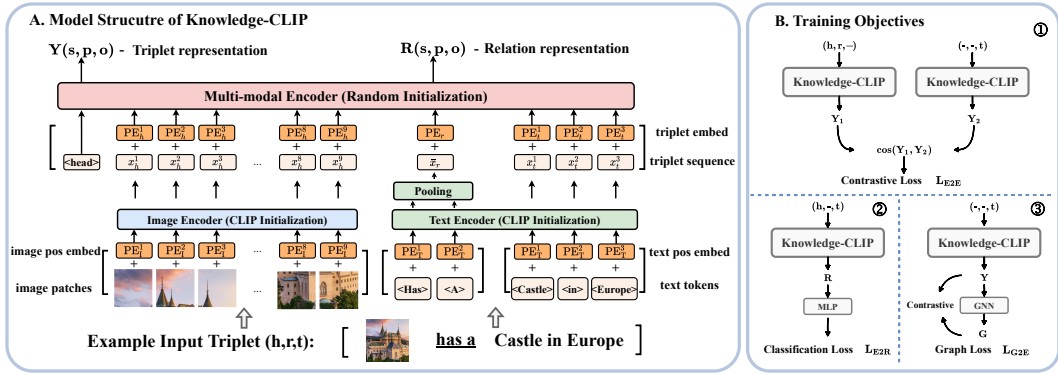

Figure 2: An overview of our framework. (A) Given a data triplet $h, r, t$ with entities $h, t$ and their relation $r$, image and text encoders first extract raw features, then a multi-modal encoder consumes the concatenated triplet sequence and outputs triplet and relation representations. (B) Three types of training objectives adopted in our framework.

CLIP uses two separate models for image encoder and text encoder respectively. For text inputs, a 12-layer Transformer is adopted with 512 width and 8 attention heads. Raw texts are first converted using byte pair encoding [40] technique under a vocabulary size of 49,152. The text sequence length is capped at 76 and added by a positional encoding before being sent into the text encoder. On the other hand, CLIP has different versions of image encoder with ResNet-based and Vision Transformer-based architectures. As the following researches have demonstrated the better performances of Vision Transformer models, we only consider Transformer-based image encoders in this paper. Similar to the text input, images are first converted to patches, and added by a positional encoding. At the last stage of both encoders, a global pooling function is adopted to compress the feature map into a single feature, which serves as the representation of the whole image/text sequence. The cosine distance of the image and text features is computed as the similarity of the data pair. For training supervision, a contrastive loss is adopted to maximize the similarity of matched pairs while minimizing the similarity of unmatched pairs. Given a batch of $N$ data pairs $\{I_i, T_i\}_{i=1}^{N}$, where $I_i$ and T represents the $i_{\text{th}}$ image and text respectively, the loss function can be parameterized as:

$$L = -\frac{1}{2}\sum_{i=1}^{N}\left(\log\frac{\exp(\cos(f_I(I_i), f_T(T_i))/\tau)}{\sum_{j=1}^{N}\exp(\cos(f_I(I_i), f_T(T_j))/\tau)} + \log\frac{\exp(\cos(f_I(I_i), f_T(T_i))/\tau)}{\sum_{j=1}^{N}\exp(\cos(f_I(I_j), f_T(T_i))/\tau)}\right),$$
(1)

where $f_I$ and $f_T$ correspond to image and text encoders respectively, $\cos(\cdot)$ denotes the cosine similarity between the inputs, and $\tau$ is a learnable temperature initialized at $0.07$.

This simple training framework actually brings several concerns that need to be addressed. First, the pre-training framework fails to model the semantic information of inputs due to the simplicity of the data structure. This results in inferior performances on tasks that require reasoning ability, *e.g.*, visual question answering and visual commonsense reasoning. Second, the image and text features reside in separate spaces, which makes it difficult to model the interactions between different modalities. Third, the massive time and resource consumption in the training procedure set restrictions on performing a full pre-training schedule from scratch.

## 4 Knowledge-CLIP

As we have summarized above, there are several concerns that hinder the transferability of CLIP and potential improvements on model performances. In this paper, we propose a novel pre-training framework based on knowledge graphs, that addresses the limitation of the original CLIP model from several perspectives: (1) we introduce knowledge graphs into the training dataset where the graph-structured data and semantic relations between concepts enable the model to extract semantic features and establish semantic connection across inputs; (2) A multi-modal encoder is added on top of the current image and text encoders to fuse the features from different modalities, and model the

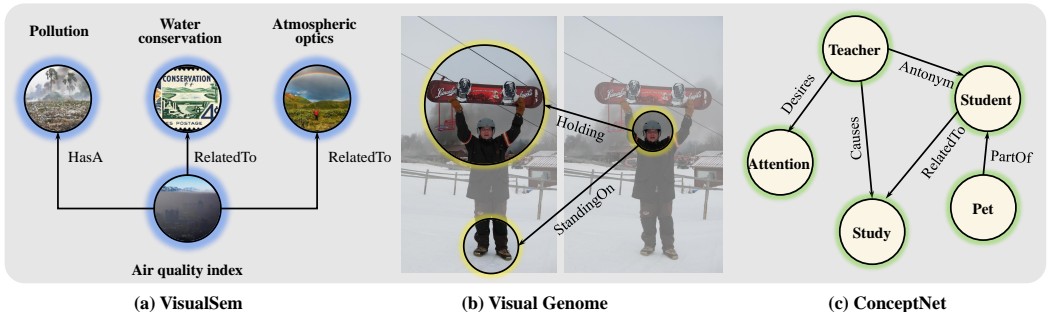

(a) VisualSem          (b) Visual Genome          (c) ConceptNet

Figure 3: Illustrations of the pre-training knowledge graph datasets, including ViusalSem [2] (multi-modal graph), Visual Genome [24] (scene graph), and ConceptNet [46] (language-based graph).

joint distribution between inputs; (3) A continuous learning strategy based on the pre-trained model of CLIP is adopted which avoids the massive computation cost in the pre-training procedure, and enhance the generalization power of the model efficiently. We introduce our framework in detail in the following sections, and show the overview in Fig. 2.

## 4.1 Data Preparation

Different from raw image-text pairs adopted in the original CLIP, our model takes knowledge graphs as input. A knowledge graph can be defined as a directed graph $\mathcal{G} = \{\xi, \mathcal{R}, \mathcal{T_R}\}$, where $\xi$, $\mathcal{R}$ correspond to sets of entities and relations, and $\mathcal{T_R}$ represent the set of relation triplets. A triplet $(h, r, t) \in \mathcal{T_R}$ denotes that entity $h \in \xi$ has relation $r \in \mathcal{R}$ with entity $t \in \xi$. As illustrated in Fig. 3, we pre-train our model on three types of knowledge graphs, including multi-modal knowledge graph, scene graph, and language-based knowledge graph. Among these, relations are constantly described in language tokens, where the entities are from different modalities in different forms.

For multi-modal knowledge graph, the entities contain both illustrative images and language descriptions. Through representing the same entity under various modalities and connecting entities with relations, it helps to build semantic connections between vision and language concepts. In practice, language and vision descriptions are randomly chosen for each entity. In this way, the triplet set $\mathcal{T_R}$ contains different forms including (Img, Rel, Img), (Img, Rel, Text), and (Text, Rel, Text), providing rich information across modalities while also enhancing perceptions within modalities.

Different from multi-modal knowledge graph, scene graph extracts visual concepts (mainly objects) for each image, and connects them with predefined semantic relations describing relative locations, actions, etc. Therefore, the entities in the scene graph correspond to a certain region in an image, with the triplet form of (Img, Rel, Img). We practically use the selected regions as the input and discard the irrelevant parts. As two entities in the same triplet denote different regions in the same image, it forces the model to extract more fine-grained features.

Lastly, language-based knowledge graph connects words and phrases of natural language with labeled edges. It is built on only language modality with the triplet form of (Text, Rel, Text), while helping to build semantic alignment within word tokens.

## 4.2 Model Architecture

The model architecture and the training framework are illustrated in Fig. 2(A). Specifically, we first process the inputs into token sequences with modality-specific tokenizers. The BPE tokenzier [40] is adopted for language inputs, while image inputs are sliced into non-overlapped patches and converted into a sequence of patches following ViT [12]. For convenient processing, we set the length of the image sequence and text sequence as $l_\text{I}$ and $l_\text{T}$ respectively for all inputs. To preserve the relative position information in the input, learnable positional encodings are added to the corresponding sequences before being sent to the model.

Two separate image encoder $f_\text{I}(\cdot)$ and text encoder $f_\text{T}(\cdot)$ are then adopted to extract features from raw inputs. For a given triplet $(h, r, t)$, the entities $h$ and $t$ are sent to the encoders with respect to

their modalities (image or text). The relation $r$, which is represented by language tokens, is sent to text encoder similar to text entity.

Compared to the model structure in CLIP, we introduce a modification to better adapt our framework. Specifically, vanilla CLIP models use a pooling function at the last layer of two encoders to compress the feature map into a global representation. Namely, for an input $u \in \mathcal{R}^{L \times d_i}$, where $L$ and $d_i$ denote the sequence length and feature dimension, the output of the encoder can be formulated as:

$$x_u = f(u) \in \mathcal{R}^{L \times d_o}, \ \ \bar{x}_u = \text{Pool}(x_u) \in \mathcal{R}^{d_o}, \tag{2}$$

where $f$ represents the feature extraction module, $\text{Pool}(\cdot)$ denotes the pooling function, and $d_o$ is the output dimension. Though efficient, it also leads to inevitable information loss in the local region, especially for the image inputs. Therefore, we remove the pooling functions for image and text entities to preserve the local information, and use $x_u \in \mathcal{R}^{L \times d_o}$ as the extracted feature. The relation, on the other hand, is normally under a limited sequence length, *e.g.*, one or two word tokens, where the information density is smaller than entities. Therefore, we retain the pooling function for relation input and use $\bar{x}_u \in \mathcal{R}^{d_o}$ as the extracted features.

In this way, we have extracted the features defined as $(x_h, \bar{x}_r, x_t)$, which correspond to the elements in the input triplet $(h, r, t)$. To model the joint distribution of different elements in the triplet, we consider a multi-modal encoder $\text{TransEncoder}(\cdot)$ to fuse the features from different sources. Specifically, we first concatenate all the features in the triplet into a single sequence and use a head token $<\text{head}>$ at the beginning of the sequence. To emphasize the status of the tokens in the sequence, we consider additional learnable encodings for each element $h, r, t$ in the triplet:

$$X(h, r, t) = [<\text{head}>, \ x_h + \text{PE}_h, \ \bar{x}_r + \text{PE}_r, \ x_t + \text{PE}_t]. \tag{3}$$

After processing by the multi-modal encoder, the feature of the head token $<\text{head}>$ finally serves as the representation of the whole sequence:

$$Y(h, r, t) = \text{TransEncoder}(X(h, r, t))[0, :]. \tag{4}$$

Also, representation for relation is extracted from the corresponding token:

$$R(h, r, t) = \text{TransEncoder}(X(h, r, t))[1 + \text{len}(x_h), :]. \tag{5}$$

## 4.3 Training Targets

Considering the unique data structure of knowledge graphs, we mainly adopt two types of training targets in our framework, including triplet-based loss and graph-based loss as illustrated in Fig. 2(B). Besides, a knowledge distillation loss is also considered due to the continuous learning strategy adopted in our framework.

**Triplet-based loss** considers a batch of triplets as the input and supervises the training of our model by estimating the joint distribution of elements in the triplets. Inspired by the mask prediction technique that models the distribution of masked tokens conditioned on the unmasked regions, we similarly mask the elements in the triplets and predict the distribution with the help of a multi-modal encoder. Specifically, for incomplete triplets where certain elements are missing in the input, the concatenated sequence can be similarly derived as in Eq. 3 by masking the corresponding feature. For example, the concatenated sequence for an input $(h, r, -)$ can be represented as:

$$X(h, r, -) = [<\text{head}>, \ x_h + \text{PE}_h, \ \bar{x}_r + \text{PE}_r, \ \mathbf{0}]. \tag{6}$$

On this basis, given a set of input $D = \{(h_i, r_i, t_i)\}_{i=1}^{N}$, we first model the distribution when one of the entities, *i.e.*, $t_i$, is masked, and derive the Entity-Entity (E2E) Loss by minimizing the negative log-likelihood:

$$-E_{(h, r) \sim D} \log(P(x_t | x_h, \bar{x}_r)). \tag{7}$$

We practically approximate the distribution $P(x_t | x_h, \bar{x}_r)$ as the cosine similarity of $P(x_t)$ and $P(x_h, \bar{x}_r)$, and defined the loss function as:

$$L_{\text{E2E}} = -\sum_{i=1}^{N} \log(\frac{\exp(\cos(Y(-, -, t_i), Y(h_i, r_i, -))/\tau)}{\sum_j \exp(\cos(Y(-, -, t_i), Y(h_j, r_j, -))/\tau)}). \tag{8}$$

We also model the distribution when the relation in the triplet is masked, and similarly derive the Entity-Relation (E2R) Loss:

$$-E_{(h,t)\sim D}\log(P(\bar{x}_r|x_h, x_t)). \tag{9}$$

Different from E2E loss, the relations in the triplets are defined in a limited set of relation groups. Therefore, we instead extract the representation of relation through an auxiliary two-layer MLP network, and model the objective as a classification problem from a predefined set of relation labels $\mathcal{R}$. In this way, the loss function can be defined as:

$$L_{\text{E2R}} = -\sum_{i=1}^{N}\sum_{r\in\mathcal{R}}\mathbf{1}_{(r=r_i)}\log(y(\bar{x}_{r_i})), \ \ \text{where} \ \ y(\bar{x}_{r_i}) = \text{MLP}(R(h_i, \text{-}, t_i)), \tag{10}$$

is extracted from an MLP model followed by the output of multi-modal encoder defined in Eq. (5).

**Graph-based loss.** We also take advantage of the graph structure in knowledge graph datasets, and adopt a graph neural network to extract deeper structural information among entities. We propagate information through connected edges in the graph, and update entity representations with aggregated feature. Specifically, for a graph neural network with $L$ layers, the update function for the $l_{\text{th}}$ layer can be formulated as:

$$G^{(l)}(t) = E_{\{h_i, r_i, t\}\in\mathcal{T}_{\mathcal{R}}} \ g^{(l-1)}(R(h_i, \text{-}, t))G^{(l-1)}(h_i), \ \ G^0(t) = Y(\text{-}, \text{-}, t), \tag{11}$$

$$\text{where} \ \ g^{(l)}(R(h_i, \text{-}, t)) = W^{(l)}R(h_i, \text{-}, t), \tag{12}$$

calculates the aggregation weights by relation representation $R(h_i, \text{-}, t)$ with a learnable matrix $W^{(l)}$.

Finally, we define the Graph-Entity(G2E) Loss by computing the cosine similarity of entity features before and after the propagation procedure in the graph:

$$L_{\text{G2E}} = -\frac{1}{\mathcal{N}_\xi}\sum_{t_i\in\xi}\log(\frac{\exp(\cos(Y(\text{-}, \text{-}, t_i), G^{(L)}(t_i))/\tau)}{\sum_{t_j}\exp(\cos(Y(\text{-}, \text{-}, t_i), G^{(L)}(t_j))/\tau)}). \tag{13}$$

**Continuous Learning.** Large-scale pre-training usually requires massive computation resources which makes it highly inefficient when training from scratch. Therefore, to inject the semantic information in an efficient manner, we consider training our model based on the pre-trained weights from the original CLIP. This powerful initialization promotes the convergence of our model and greatly enhances the training efficiency. However, naively extending the training process with new data leads to severe forgetting problem that hampers the performance of the original models.

To address this limitation, we adopt simple solutions to maintain CLIP performances while improving its ability to extract semantic features from knowledge graphs. (1) Besides the knowledge graph datasets, we also train our model on several widely adopted image-text datasets that share a similar data distribution with the training data in CLIP. To better fit our pre-training framework, we convert the original image-text pair into the form of triplets, with specifically designed relations 'image of' and 'caption of'. (2) We also use the original CLIP model as the teacher, and use an auxiliary loss $L_{\text{KD}}$ to measure the KL distance between the output of CLIP and our model.

Overall, the final pre-training objective of Knowledge-CLIP is formulated as:

$$L = L_{\text{E2E}} + L_{\text{E2R}} + L_{\text{G2E}} + L_{\text{KD}}. \tag{14}$$

## 5 Experiments

### 5.1 Implementation Details

**Experimental Setup.** In all the experiments, we use the same model structure as CLIP [38]. A 12-layer Transformer model with 512 width is adopted for text encoder, and ViT-L/14 is adopted for image encoder. For text and image encoder, we use the pre-trained weights in the original CLIP as the initialization. For the multi-modal encoder, we consider a 4 layer Transformer model with 1024 width. The rate for drop path is set as 0.1 during training. As the added multi-modal encoder is trained from random initialization, we decrease the learning rate for the pre-trained weights from CLIP to achieve a more balanced step in the optimization. We train Knowledge-CLIP with an initial

Table 1: Fine-tuned image-text retrieval results on Flockr30K and COCO datasets. The best result is shown in blue and the better result between CLIP and our approach is shown in **bold**.

| Method | Flickr30K (1K test set) | | | | | | MSCOCO(5K test set) | | | | | |
| | Text Retrieval | | | Image Retrieval | | | Text Retrieval | | | Image Retrieval | | |
| | R@1 | R@5 | R@10 | R@1 | R@5 | R@10 | R@1 | R@5 | R@10 | R@1 | R@5 | R@10 |
|---|---|---|---|---|---|---|---|---|---|---|---|---|
| UNITER [9] | 87.3 | 98.0 | 99.2 | 75.6 | 94.1 | 96.8 | 65.7 | 88.6 | 93.8 | 52.9 | 79.9 | 88.0 |
| VILLA [14] | 87.9 | 97.5 | 98.8 | 76.3 | 94.2 | 96.8 | - | - | - | - | - | - |
| OSCAR [28] | - | - | - | - | - | - | 73.5 | 92.2 | 96.0 | 57.5 | 82.8 | 89.8 |
| ERNIE-ViL [67] | 88.7 | 98.0 | 99.2 | 76.7 | 93.6 | 96.4 | - | - | - | - | - | - |
| Unicoder-VL [25] | 86.2 | 96.3 | 99.0 | 71.5 | 91.2 | 95.2 | 62.3 | 87.1 | 92.8 | 48.4 | 76.7 | 85.9 |
| ViLT [23] | 83.5 | 96.7 | 98.6 | 64.4 | 88.7 | 93.8 | 61.5 | 86.3 | 92.7 | 42.7 | 72.9 | 83.1 |
| Uni-Perceiver [72] | 87.9 | 98.2 | 99.1 | 74.9 | 93.5 | 96.0 | 64.7 | 87.8 | 93.7 | 48.3 | 75.9 | 84.5 |
| CLIP [38] | 88.6 | 98.5 | **99.4** | 72.4 | 92.3 | 96.6 | 67.3 | 85.4 | 92.4 | 54.3 | 83.5 | 90.0 |
| **Ours** | **89.2** | **98.9** | **99.4** | **75.7** | **94.4** | **96.8** | **70.2** | **89.2** | **94.4** | **57.6** | **83.9** | **90.4** |

learning rate of 1e-5 for image and text encoders, and 1e-3 for the multi-modal encoder. Cosine learning rate with linear warmup is used in the training schedule. Weight decay and gradient clip are also adopted. See more details in the supplemental material.

**Pre-train Dataset.** Three knowledge graph datasets are adopted in the pre-training process. VisualSem [2] is a high-quality multi-modal knowledge graph dataset for vision and language concepts, including entities with multilingual glosses, multiple illustrative images, and visually relevant relations, covering a total number of 90k nodes, 1.3M glosses and 938k images. 13 semantic relations are used to connect different entities in the graph, while the entities in VisualSem are linked to Wikipedia articles, WordNet [34], and high-quality images from ImageNet [10]. Visual Genome [24] is a knowledge-based scene graph dataset that connects structured image concepts with semantic relations. Visual Genome serves as the benchmark for various vision tasks, *e.g.*, visual grounding, and scene graph generation. ConceptNet [46] is a knowledge graph that connects words and phrases of natural language with labeled edges. Its knowledge is collected from many sources including expert-created resources and crowd-sourcing built on only language modality.

Besides the three knowledge graph datasets, we also train our model on two widely adopted image-text datasets that share the similar data distribution with the training data in CLIP. We practically add COCO Caption [8] and CC3M [42] to the training set, while large-scale datasets like CC12M [4] or YFCC [21] are not considered to maintain training efficiency.

**Downstream Task.** To validate the effectiveness of our framework, we conduct experiments on various downstream tasks, including multi-modal tasks like text and image retrieval, visual question answering, and uni-modal tasks like image classification and natural language understanding.

## 5.2 Multi-modal Tasks

**Visual question answering / Visual Entailment.** We also validate the effectiveness of Knowledge-CLIP on other vision-language tasks, including VQA [15] and SNLI-VE [64]. We show the comparison results in Tab. 2. Compared to competitive baselines including VILLA [14] and ALBEF [26], Knowledge-CLIP with ViT-L/14 shows better performances under all settings, while the smaller model also achieves competitive re-

Table 2: Fine-tuned results on other V-L tasks.

| Method | VQA | | SNLI_VE | |
| | test-dev | test-std | val | test |
|---|---|---|---|---|
| UNITER [9] | 72.70 | 72.91 | 78.59 | 78.28 |
| VILLA [14] | 73.59 | 73.67 | 79.47 | 79.03 |
| OSCAR [28] | 73.16 | 73.44 | - | - |
| ALBEF [26] | 74.54 | 74.70 | 80.14 | 80.30 |
| Uni-Perceiver [72] | 73.4 | 74.1 | - | - |
| FLAVA [45] | 72.8 | - | 78.89 | - |
| CLIP [38] | 74.10 | 73.56 | 79.51 | 80.01 |
| **Ours** | **76.11** | **75.24** | **80.52** | **80.97** |

sults. Compared to the original CLIP model, our pre-trained model practically improves its transferability on downstream tasks, especially on the datasets like VQA that requires reasoning ability.

**Image and text retrieval.** We first conduct experiments on Flickr30k [37] and COCO Caption [8] dataset to show the performances of our model on image-text retrieval tasks. Given input sets $\mathcal{X}$

Table 4: Fine-tuned language understanding results on GLUE dataset. The best result is shown in blue and the better result between CLIP and our approach is shown in **bold**.

| Method | CoLA Mcc. | SST-2 Acc. | RTE Acc. | MRPC Acc./F1 | QQP Acc./F1 | MNLI Acc | QNLI Acc |
|--------|-----------|------------|----------|--------------|-------------|----------|----------|
| VilBERT [32] | 36.1 | 90.4 | 53.7 | 69.0/79.4 | 88.6/85.0 | 79.9 | 83.8 |
| VL-BERT [48] | 38.7 | 89.8 | 55.7 | 70.6/81.8 | 89.0/85.4 | 81.2 | 86.3 |
| UNITER [9] | 37.4 | 89.7 | 55.6 | 69.3/80.3 | 89.2/85.7 | 80.9 | 86.0 |
| SimVLM [60] | 46.7 | 90.0 | 63.9 | 75.2/84.4 | 90.4/87.2 | 83.4 | 88.6 |
| FLAVA [45] | 50.7 | 90.9 | 57.8 | 81.4/86.9 | 90.4/87.2 | 80.3 | 87.3 |
| CLIP [38] | 42.1 | 90.5 | 59.2 | 82.4/87.0 | 90.4/87.1 | 80.9 | 87.1 |
| **Ours** | **50.4** | **91.2** | **62.4** | **83.5/87.6** | **90.5/87.9** | **83.6** | **89.5** |

and $\mathcal{Y}$ of images and texts, we use Knowledge-CLIP to extract features for each input, and model the joint probability with the cosine similarity between image and text pairs. We summarize the comparison results of Knowledge-CLIP with competitive baselines in Tab. 1. It is shown that our model consistently achieves better results over the original CLIP on both datasets, while comparable with competitive baselines like OSCAR.

## 5.3 Uni-modal Tasks

**Image Classification.** To further demonstrate the generalization power of Knowledge-CLIP, we compare the performances of pre-train models on the ImageNet classification task [10]. We summarize the comparison results in Tab. 3, and show that Knowledge-CLIP can also handle vision tasks well. We argue the improvements over baselines may attribute to the scene graphs in our pre-training dataset, which emphasize the visual concepts in the images.

Table 3: Fine-tuned results on ImageNet.

| Method | Acc(%) |
|--------|--------|
| DeiT [53] | 83.4 |
| CLIP [38] | 84.2 |
| **Ours** | **84.4** |

**Language Understanding.** We validate the generalization performance of Knowledge-CLIP for language understanding tasks on the widely adopted GLUE dataset [55]. Specifically, we conduct experiments on 7 tasks in GLUE and summarize the comparison results in Tab. 4. It is shown that our model achieves comparable performances with competitive baseline models. Also, for tasks like QQP and MNLI that require sentence-pair matching, Knowledge-CLIP shows higher performances, due to the existence of language triplets in the pre-training dataset.

## 5.4 Ablation Studies

To validate the effectiveness of the components in our work, we carefully design several settings, including (1) CLIP+continuous learning: we train vanilla CLIP (pretrained weights as initialization) on knowledge datasets adopted in our work; (2) Knowledge-CLIP-(t1, t2, t3): we remove the training objectives respectively in our work to analyze the contribution of each loss. For all experiments, we adopt a smaller model (ViT-B/32) as the image encoder of CLIP in the ablation study. Also, it is worth noticing that KD loss plays a vital role in the continuous learning scheme, without which will lead to a significant performance drop due to the model forgetting problem. Therefore, we use KD loss in all the ablation settings for a fair comparison.

We show the comparison results on two representative tasks in Tab. 5, including the image/text retrieval task on Flickr30K, and the visual question answering task in VQA. Several observations can be made from the ablation: (1) All three training objectives (E2E, E2R, G2E) contribute to improving the model performance. Training the model without any of the objectives leads to inferior performances on downstream tasks. We argue that the E2E, E2R, and G2E loss promote the model from different perspectives by focusing on semantic understanding of concepts, complicated relations between entities, and structural information. Therefore, all three objectives are necessary for the framework and contribute to the improvement respectively. (2) By comparing the first and second row, we can see that simply training the CLIP model with extra time and data fails to improve the generalization performance. It also demonstrates that the improvements mainly come from the injected knowledge information rather than the continuous learning scheme.

Table 5: Ablation studies of continuous learning / training objectives. We report results on Flickr30K retrieval task and VQA task with ViT-B/32 as image encoder.

| Model | KG data | E2E | E2R | G2E | Flickr30K retrieval | | VQA | |
| --- | --- | --- | --- | --- | --- | --- | --- | --- |
| | | | | | Text (R@1) | Image (R@1) | test-dev | test-std |
| CLIP | - | - | - | - | 84.2 | 63.1 | 68.9 | 69.2 |
| CLIP+Continuous Learning | ✓ | - | - | - | 84.5 | 63.0 | 69.1 | 69.5 |
| Knowledge-CLIP-t1 | ✓ | - | ✓ | ✓ | 85.0 | 64.6 | 70.4 | 71.1 |
| Knowledge-CLIP-t2 | ✓ | ✓ | - | ✓ | 85.7 | 66.0 | 71.2 | 69.9 |
| Knowledge-CLIP-t3 | ✓ | ✓ | ✓ | - | 84.9 | 65.8 | 70.2 | 70.4 |
| Knowledge-CLIP (Full) | ✓ | ✓ | ✓ | ✓ | **86.3** | **67.2** | **72.5** | **72.7** |

We also conduct an ablation study on the KD loss adopted for continuous learning and summarize the results in Tab. 6. The model achieves lower results after removing the KD loss, indicating its vital role in the continuous learning scheme. We argue the reason for this phenomenon is that the model suffers from the forgetting problem, which is widely spotted in the field of lifelong learning and continuous learning.

Table 6: Ablation studies of the KD loss.

| Model | Flickr30K retrieval | |
| --- | --- | --- |
| | Text (R@1) | Image (R@1) |
| Ours w/o KD | 82.4 | 62.5 |
| **Ours w/ KD** | **86.3** | **67.2** |

## 5.5 Analysis on particular semantics

We also conduct experiments on carefully selected data which may better reflect how a vision-language model understands a particular type of input. Specifically, we select questions in the VQA dataset that contains (1) Negations; (2) Color attributes; (3) Position attributes; (4) Sizes. We summarize the comparison results of CLIP and our model on these sub-datasets in Tab. 7.

As we can observe, our model achieves consistent improvements over CLIP on these specially designed datasets and shows significantly better results. Regarding questions with negation, our model achieves 2.1% higher accuracy. Regarding color and position attributes, our model shows even higher improvements. We believe these comparisons on different 'semantic domains' demonstrate

Table 7: Ablation studies on semantic inputs.

| Dataset | CLIP | Knowledge-CLIP |
| --- | --- | --- |
| Negation | 64.7 | **66.8**(+2.1) |
| Color | 54.2 | **59.9**(+5.7) |
| Position | 61.2 | **68.3**(+7.1) |
| Size | 72.1 | **63.4**(+1.3) |

the effectiveness of injecting knowledge information into the current vision-language pretraining framework which practically enhances the model perception of semantic understanding.

## 6 Conclusion

In this paper, we propose a novel vision-language pretraining framework that incorporates knowledge information to model the semantic connections between vision and language entities. We introduce three types of graph-structured datasets into the training process, and adopt a multi-modal encoder to model the joint distribution of entities and their semantic relations. Extensive experiments on various downstream tasks including multi-modal, uni-modal, and graph-based tasks validate the transfer and generalization ability of our model. Our approach is now limited in injecting knowledge information into the CLIP models. However, our training objectives and new knowledge graph datasets are technically compatible with other large-scale pretraining frameworks. We will explore the possibility of further applications in the future.

## 7 Acknowledgement

This work is supported in part by the National Key R&D Program of China under Grant 2020AAA0105200, the National Natural Science Foundation of China under Grants 62022048, Guoqiang Institute of Tsinghua University and Beijing Academy of Artificial Intelligence. We also appreciate the generous donation of computing resources by High-Flyer AI.

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
