# Contrastive Language-Image Pre-Training with Knowledge Graphs – Supplementary Material

## A. Implementation Detail

### A.1 Pre-training

Our model is trained on three knowledge graph datasets and two image-text datasets whose statistics are listed in Tab. 1.

Table 1: Pre-training dataset statistics.

| Dataset | #Images | #Text | #Triplets | #Relation Classes |
|---------|---------|-------|-----------|-------------------|
| VisualSem [1] | 90K | 1.3M | 90K | 13 |
| Visual Genome [3] | 108K | - | 540K | 50 |
| ConceptNet [6] | - | 180K | 249K | 17 |
| CC3M [5] | 3.0M | 3.0M | 3.0M | 2 |
| COCO Caption [2] | 113K | 567K | 567K | 2 |

For CC3M and COCO Caption, we convert the original image-text pairs to triplets by adding self-defined semantic relations 'image of' and 'caption of'. 75% data in a training batch are sampled from knowledge graph datasets, and the rest are sampled from image-text datasets. We set the maximum length of image/text inputs as $l_I = 256$ and $l_T = 77$ respectively for convenient processing. Weight decay is set as 0.05 and gradient clip is set as 5.

For data processing, the five datasets we used are all public datasets that have been widely used in early works. Therefore, we practically follow the data processing routine. Specifically, for VisualSem, each concept (entity) in the triplet has both corresponding images and text descriptions and will be randomly chosen if the triplet is sampled. In this way, the modality of the concept in different triplets or training batches can be different, and the triplet forms can include image/text, relation, image/text. Differently, the Visual Genome dataset contains scene graphs for each image. The nodes are presented in a bounding box and the edges are represented by word tokens, e.g., standing on. We extract the image features of the corresponding box and generate image, relation, image triplets. For each image in Visual Genome, we randomly sample 4 triplets, based on the consideration that a larger number may lead to repeated sampling. The triplets in ConceptNet are pre-processed and explicitly given by the authors. So we directly sample them in the training batch. For CC3M and COCO Caption, we convert the original image-text pairs to triplets by adding self-defined semantic relations 'image of' and 'caption of'.

For each input modality in the training data, we adopt a unified processing procedure to make it possible for batch training. Specifically, the length of the image is set as 16x16 and the length of the text is set as 77. We adopt the same data augmentation as vanilla CLIP including resize, center crop, and normalization for images. For text, a start of text token and an end of text token are first concatenated with the input and the BPE tokenizer is adopted to encode the words. For each training batch, 75% of data is sampled from the three knowledge graph datasets, and 25% of data is sampled from CC3M/COCO Caption.

When computing the G2E loss, we actually construct small graphs/sub-graphs. Specifically, for the multi-modal dataset VisualSem and text knowledge graph dataset ConceptNet, only triplets are given in the original dataset. Therefore, we generate graphs by first sampling a center node and growing the graph within two-hop neighbors. We further constrain the number of one-hop neighbors to be smaller than 4 to control the scale of the generated graphs. For the scene graph dataset Visual Genome, a

Submitted to 36th Conference on Neural Information Processing Systems (NeurIPS 2022). Do not distribute.

scene graph is naturally provided for each image. In this case, we gradually prune the graph to a sub-graph until satisfying the aforementioned demand for the other two datasets.

## A.2 Fine-tuning

**Image and text retrieval.** Image and text retrieval take image and text $\{I, T\}$ as input separately, and predict the corresponding feature $Y(I, \text{-}, \text{-})$ and $Y(\text{-}, \text{-}, T)$. Then, the most similar image and text features serve as the retrieved output.

**VQA** task takes image-text pair as input, and requires the model to provide the corresponding answer. The question is given in the image-text pair, and the model is expected to provide the answer. Usually, candidate answers are provided in language descriptions. Specifically, given a image and question $\{I, Q\}$, and given an answer A, the model predicts the features of $Y(I, \text{-}, Q)$ and $Y(A, \text{-}, \text{-})$ and use the most similar features as the answer to the given question Q.

**VE** task is similar to VQA, which also takes image-text pair as input. Differently, the model is expected to classify if the given text correctly describes the image (Entailment), does not describe the image (Contradiction), or hard to tell (Neutral). We practically convert the candidate answers into 'yes', 'no', and 'neutral' to represent the original answers. The prediction process is the same as VQA.

**Image Classification** is performed following the setup in CLIP [4]. We extract the feature of image I as $Y(I, \text{-}, \text{-})$ and extract the target labels in a template T as $Y(\text{-}, \text{-}, T)$. The prediction process is the same as retrieval tasks, where the most similar target label is served as the final prediction.

**Language Understanding** is similar to retrieval tasks. Instead, it only consumes text pairs as input and chooses the most similar one as the predicted output.

## B. Additional examples for Figure 1 (main paper)

We show the comparison between vanilla CLIP and our methods on the toy examples shown in Figure 1 of the main paper. It can be observed in Fig. 1 that by injecting knowledge information, model perception towards these semantic descriptions is promoted.

To better illustrate our claim, we give two additional toy examples to show how the vanilla CLIP model handles semantic inputs. The first example shown in Fig. 2 contains an image with two main objects: a white car and a red house. In this case, we consider two templates including 'a photo of a white {}' and 'a photo of a red {}'. It is shown vanilla CLIP still tends to provide similar outputs and recognize the same object in the image. This proves that vanilla CLIP fails to understand the meaning of color descriptions.

The second example shown in Fig. 3 considers scenarios with size and location descriptions. Given a photo of a strawberry and an apple, we use the template of 'a photo of small {} and big {}' and 'a photo of {} on the left and {} on the right' as the input. In this case, we constrain the candidate text token to {apple, strawberry} to better reflect the model bias. As a result, CLIP also fails to understand the semantic meaning and recognizes the relative position/scale of the objects.

We believe the aforementioned examples can help support our claim that the image-text training scheme in CLIP fails to provide semantic perceptions, and injecting knowledge information may be a feasible direction. We also provide the prediction of our method in these examples and show that a knowledge-based training scheme can practically help model perception on these semantic descriptions.

## C. Visualization results on downstream tasks

We show comparison results on downstream tasks including retrieval and vqa tasks in Fig. 4 and Fig. 5.

**(a.1) CLIP predictions on templates with opposite semantic description**

**(a.2) Knowledge-CLIP predictions on templates with opposite semantic description**

**(b.1) CLIP predictions templates with wrong semantic description**

**(b.2) Knowledge-CLIP predictions templates with wrong semantic description**

Figure 1: Comparison between CLIP and Knowledge-CLIP with opposite semantic descriptions, e.g., adding 'not' in the template or describing an image with wrong color. Best view in color.

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

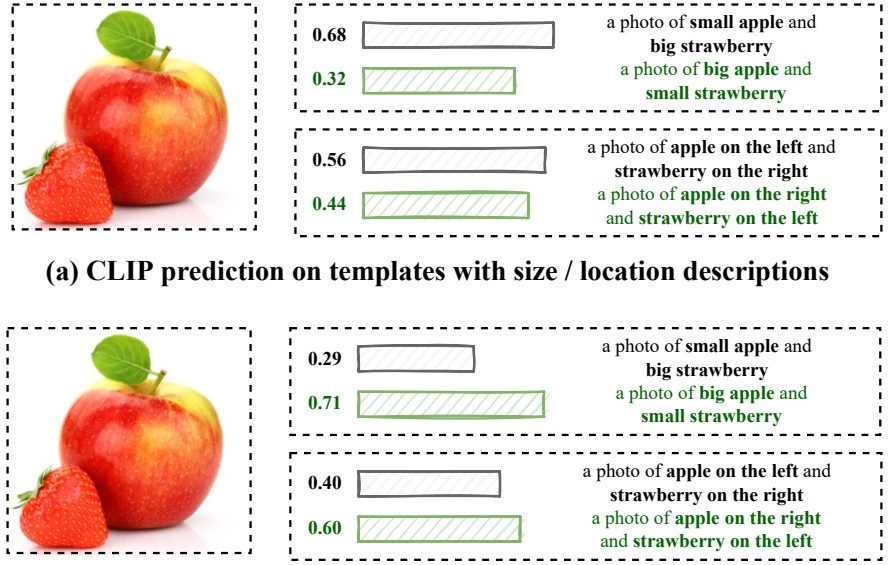

template: a photo of **white**{}      template: a photo of **red** {}

| | |
|---|---|
| 0.63 tile roof | 0.69 tile roof |
| 0.07 station wagon | 0.13 solar thermal collector |
| 0.05 mobile home | 0.04 station wagon |
| 0.04 solar thermal collector | 0.03 mobile home |
| 0.02 convertible | 0.01 thatched roof |

**(a) CLIP prediction on templates with color descriptions**

template: a photo of **white**{}      template: a photo of **red** {}

| | |
|---|---|
| 0.21 station wagon | 0.56 tile roof |
| 0.18 tile roof | 0.03 mobile home |
| 0.11 solar thermal collector | 0.03 station wagon |
| 0.06 convertible | 0.02 solar thermal collector |
| 0.05 sports car | 0.01 convertible |

**(b) Knowledge-CLIP prediction on templates with color descriptions**

Figure 2: Comparison between CLIP and Knowledge-CLIP with different color descriptions. Better view in color.

0.68   a photo of **small apple** and **big strawberry**

0.32   a photo of **big apple** and **small strawberry**

0.56   a photo of **apple on the left** and **strawberry on the right**

0.44   a photo of **apple on the right** and **strawberry on the left**

**(a) CLIP prediction on templates with size / location descriptions**

0.29   a photo of **small apple** and **big strawberry**

0.71   a photo of **big apple** and **small strawberry**

0.40   a photo of **apple on the left** and **strawberry on the right**

0.60   a photo of **apple on the right** and **strawberry on the left**

**(b) Knowledge-CLIP prediction on templates with size / location descriptions**

Figure 3: Comparison between CLIP and Knowledge-CLIP with different scale / location descriptions. Correct answers are shown in green. Better view in color.

[5] Piyush Sharma, Nan Ding, Sebastian Goodman, and Radu Soricut. Conceptual captions: A cleaned, hypernymed, image alt-text dataset for automatic image captioning. In *Proceedings of the 56th Annual Meeting of the Association for Computational Linguistics (Volume 1: Long Papers)*, pages 2556–2565, 2018. 1

[6] Robyn Speer, Joshua Chin, and Catherine Havasi. Conceptnet 5.5: An open multilingual graph of general knowledge. In *Thirty-first AAAI conference on artificial intelligence*, 2017. 1

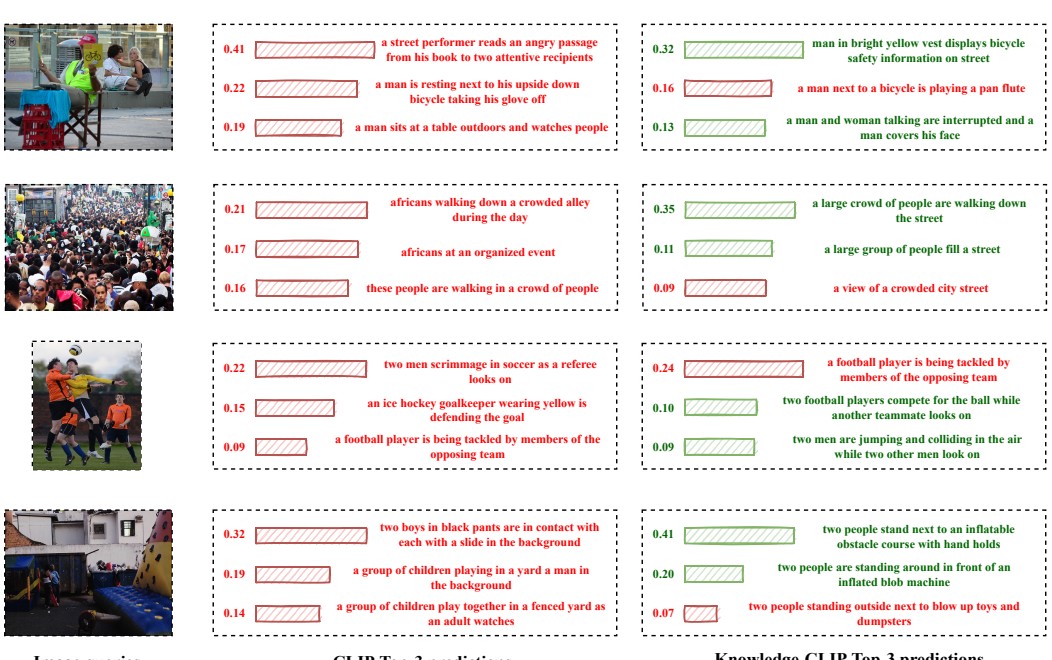

**Image queries**  **CLIP Top-3 predictions**  **Knowledge-CLIP Top-3 predictions**

Figure 4: Visualization results on retrieval tasks. Correct answers are shown in green and wrong answers are shown in red. Better view in color.

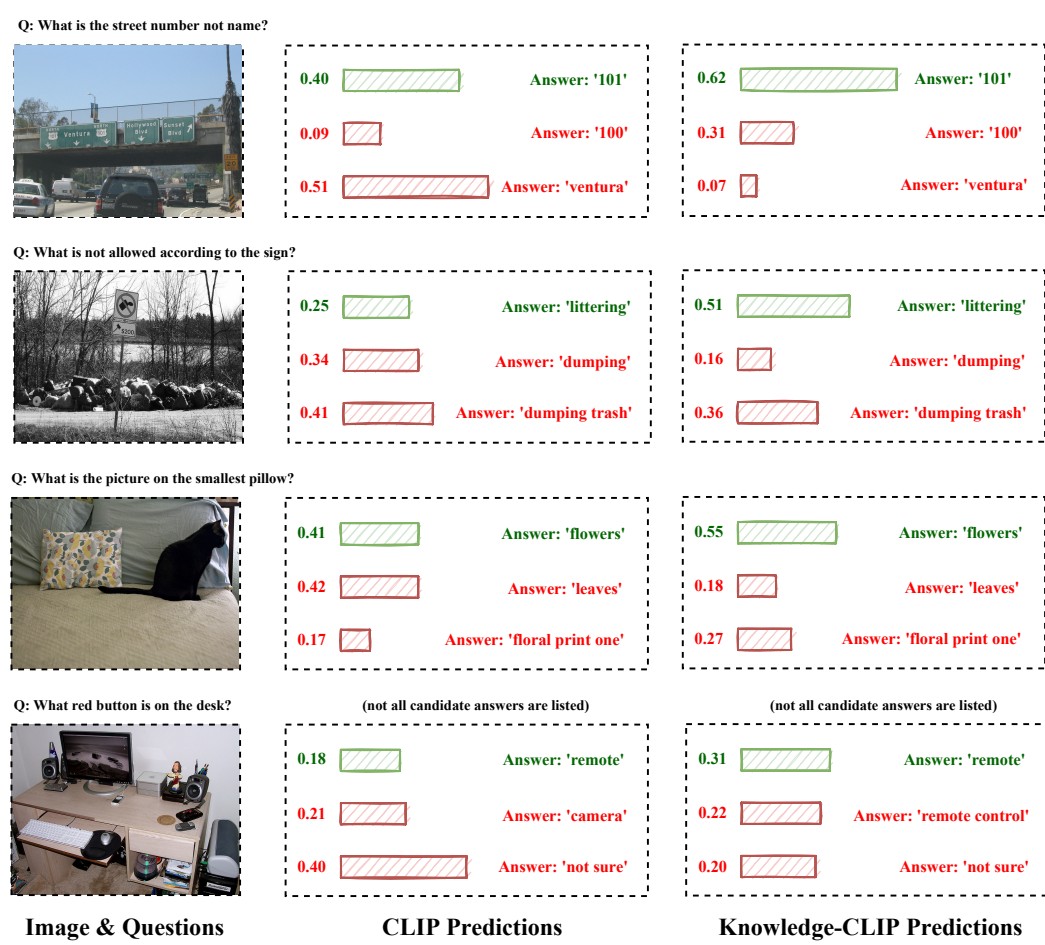

**Image & Questions**   **CLIP Predictions**   **Knowledge-CLIP Predictions**

Figure 5: Visualization results on VQA task. Correct answers are shown in green and wrong answers are shown in red. Better view in color.