# OpenReview forum: "Contrastive Language-Image Pre-Training with Knowledge Graphs"
_NeurIPS.cc/2022/Conference — NeurIPS 2022 Accept_

### Official Review · Reviewer_Foeh · 2022-07-09

**Rating:** 6
**Confidence:** 4
**Soundness:** 3 good
**Presentation:** 3 good
**Contribution:** 3 good

**Summary:**

- The paper proposes a new Contrastive Language-Image Pre-Training method that incorporates semantic knowledge graphs (knowledge-CLIP). The method builds on the existing CLIP architecture and introduces three KG-aware pretraining objectives on top: Entity-Entity loss, Entity-Relation loss, Graph-Entity loss. The authors experiment with this method using multiple knowledge graphs and show improved performance across multimodal, unimodal, and KG-based downstream tasks.

**Questions:**

- Please find my main questions and suggestions mentioned in the "Weaknesses" section above.
- Will the code/data/trained models be released?
- For Figure 1b, what is the expected prediction when changing "yellow" to "green"? I wonder "a photo of green __" may not really make sense for the given image because there is no green object in the image?
- For graph-based loss (Equation 11), do you compute Y(-,-,t) for all the triplets using knowledge-CLIP? Would  that be very expensive? Also, what is the size of the graph considered in the GNN - is it the entire KG or some subgraph?
- A suggestion for the presentation: In Figure 2, maybe make the font consistent across A (left panel) and B (right panel)?

**Limitations:**

- The authors addressed the potential negative societal impact (in Appendix)

**Strengths And Weaknesses:**

Strengths
- The paper is generally well-written and easy to understand.
- The problem they are solving (incorporating background semantic knowledge into language-image model) and their proposed method (KG-aware model architecture and training objectives) are well motivated. The idea of considering a triplet set that contains different forms, (Img, Rel, Img), (Img, Rel, Text), (Text, Rel, Text), is interesting.
- The experiments are solid (covers three types of KGs; covers both multimodal and unimodal downstream tasks) and shows that the proposed method makes moderate improvements over the baseline.

Weaknesses
- While the authors show improved numbers on benchmark datasets, it would be nice to also show and discuss how the proposed knowledge-CLIP model is *qualitatively* improving over the baseline CLIP. For example, in Intro and Figure 1, the authors motivates this paper by arguing that the baseline CLIP only captures text-image co-occurrence and fails to adjust for negation in text, etc. - is this issue solved in the proposed knowledge-CLIP model? Some existing work that combines text and KG (e.g. https://arxiv.org/abs/2104.06378) has done closely-related analyses such as adding negation or changing entities in text to see if the KG-augmented method can robustly handle them. It would be very interesting if the authors perform such analysis on the proposed knowledge-CLIP model that combines image, text and KGs.

---

> ### Author Response · Authors · 2022-08-02
> **Response to Reviewer Foeh (Part 2)**
>
> **4. Figure 1**  *what is the expected prediction when changing "yellow" to "green"? I wonder "a photo of green __" may not really make sense for the given image because there is no green object in the image?*
>
> Sorry for the confusion. We have updated several new examples in the supplementary material for better illustration.
>
> Specifically,  we give two additional toy examples in the supplementary material to show how the vanilla CLIP model handles semantic inputs. The first example (shown in Supplementary Figure 2) contains an image with two main objects: a **white** car and a **red** house. In this case, we consider two templates including 'a photo of a white {}' and 'a photo of a red {}'. Unfortunately, vanilla CLIP still tends to provide similar outputs and recognize the same object in the image. This proves that vanilla CLIP fails to understand the meaning of color descriptions.
>
> The second example (shown in Supplementary Figure 3) considers scenarios with size and location descriptions. Given a photo of a strawberry and an apple, we use the template of 'a photo of small {} and big {}' and 'a photo of {} on the left and {} on the right' as the input. In this case, we constrain the candidate text token to {apple, strawberry} to better reflect the model bias. As a result, CLIP also fails to understand the semantic meaning and recognizes the relative position/scale of the objects.
>
> We believe the aforementioned examples can help support our claim that the image-text training scheme in CLIP fails to provide semantic perceptions, and injecting knowledge information may be a feasible direction. We also provide the prediction of our method in these examples and show that a knowledge-based training scheme can practically help model perception on these semantic descriptions.
>
> We also provide several visualization results in Supplementary Figure 4 and Figure 5. The reviewer may refer to these two figures to better understand the improvements brought by our methods.
>
>
>
> **5. For graph-based loss (Equation 11), do you compute Y(-,-,t) for all the triplets using knowledge-CLIP? Would that be very expensive? Also, what is the size of the graph considered in the GNN - is it the entire KG or some subgraph?**
>
> Yes, we compute all Y(-,-,t) to derive the graph-based loss. As the reviewer has pointed out, to avoid the successive computation by large graphs, we actually construct small graph/sub-graphs. Specifically, for the multi-modal dataset VisualSem and text knowledge graph dataset ConceptNet, only triplets are given in the original dataset. Therefore, we generate graphs by first sampling a center node and growing the graph within two-hop neighbors. We further constrain the number of one-hop neighbors to be smaller than 4 to control the scale of the generated graphs. For the scene graph dataset Visual Genome, a scene graph is naturally provided for each image. In this case, we gradually prune the graph to a sub-graph until satisfying the aforementioned demand for the other two datasets. We have added the processing detail in the supplementary (Sec.O.5).
>
>
>
> **6. In Figure 2, maybe make the font consistent across A (left panel) and B (right panel)?**
>
> Thanks for the suggestion. We have revised it in the new version.
>
>
> Lastly, we are grateful for your time and we hope we have adequately addressed your concerns. If you have any other concerns, please let us know. We look forward to additional discussion.

---

> ### Author Response · Authors · 2022-08-02
> **Response to Reviewer Foeh (Part 1)**
>
> We would first like to express our appreciation for your time and insightful comments. Please find our response to your concerns in the following:
>
> **1. How the proposed knowledge-CLIP model is *qualitatively* improving over the baseline CLIP.** *For example, in Intro and Figure 1, the authors motivates this paper by arguing that the baseline CLIP only captures text-image co-occurrence and fails to adjust for negation in text, etc. - is this issue solved in the proposed knowledge-CLIP model?*
>
> Thanks for the suggestion. We have updated the prediction of our model regarding the two examples in the supplementary material. We also provide additional examples to better support our claim. The results in Figure 1-3 (supplementary) show that our method can improve the model's ability on understanding object attributes like color/scale and the relative position between objects. The results in Figure 4 (supplementary) show that our model can handle images with multiple objects, e.g., row 1 and 2, and handle actions/interactions between objects, e.g., row 3 and 4. The results in Figure 5 (supplementary) show that our model can understand the negation, e.g., rows 1 and 2, and has better reasoning ability.
>
>
>
> **2. Analysis on the proposed knowledge-CLIP** *Some existing work that combines text and KG (e.g. https://arxiv.org/abs/2104.06378) has done closely-related analyses such as adding negation or changing entities in text to see if the KG-augmented method can robustly handle them. It would be very interesting if the authors perform such analysis on the proposed knowledge-CLIP model that combines image, text and KGs.*
>
> Thanks for the valuable suggestion. We follow similar settings in QA-GNN (https://arxiv.org/abs/2104.06378) and conduct experiments on carefully selected data which may better reflect how a vision-language model understands a particular type of input. Specifically, we select questions in the VQA dataset that contains (1) Negation including: 'not' / 'no' / 'nothing' / 'unlikely' (2) Color attribute including : 'green' / 'red' / 'yellow' / 'white' (3) Position attribute: 'between' / 'behind' / 'along' / 'holding' / 'in front of / 'sitting on' / 'standing on' / 'under' (4) Size: 'small' / 'smaller' / 'smallest' / 'big' / 'bigger' / 'biggest' / 'tiny' / 'huge' / 'medium'. We summarize the comparison results of CLIP and our model on these sub-datasets below.
>
> Table 1 Ablation studies on particular type of input.
>
> | **Datasets** | **CLIP** | **Knowledge-CLIP** |
> | :----------: | :------: | :----------------: |
> |   Negation   |   64.7   |    66.8 (+2.1)     |
> |    Color     |   54.2   |    59.9 (+5.7)     |
> |   Position   |   61.2   |    68.3 (+7.1)     |
> |     Size     |   72.1   |    73.4 (+1.3)     |
>
> As we can observe, our model achieves consistent improvements over CLIP on these specially designed datasets and shows significantly better results. Regarding questions with negation, our model achieves 2.1% higher accuracy. Regarding color and position attributes, our model shows even higher improvements. We believe these comparisons on different 'semantic domains' demonstrate the effectiveness of injecting knowledge information into the current vision- language pretraining framework which practically enhances the model perception of semantic understanding.
>
> We have updated the results and discussion in the supplementary (Sec.O.4).
>
>
>
> **3. Will the code/data/trained models be released?**
>
> Yes. The dataset in our pre-training procedure is already publicly available, and we will release the training code and models if the paper is accepted.

---

> ### Comment · Reviewer_Foeh · 2022-08-09
> **Thank you for your response**
>
> Many thanks to the authors for answering the questions and providing additional analyses.
>
> - Thanks for the model prediction examples and explanation of Figure 1. I can now better understand the intuition of the proposed work
>
> - New analyses on the benefit of KG are great and address my concerns
>
> I have raised my rating.

---

### Official Review · Reviewer_WaXR · 2022-07-11

**Rating:** 5
**Confidence:** 4
**Soundness:** 3 good
**Presentation:** 3 good
**Contribution:** 3 good

**Summary:**

In order to address the semantic alignment issue in multimodal tasks, this paper suggests the Knowledge-CLIP framework. Experiments reveal that the proposed approach is practical.


**Questions:**

Please see and answer "Strengths and Weaknesses" for the issues I mentioned. Additionally, the reviewer suggests the authors take the time to consider why Multimodal Pre-training requires Graph (or knowledge).  The authors claimed that focusing on the graph can help semantic alignment at first, but the second half of the paper, just shows that continue learning with graph will have some improvement to CLIP.


**Limitations:**

I recommend the authors give more robust explanations. Besides, this article still has many self-justifying explanations so far, and I hope the author will give more clear and reasonable illustrations. Overall, this article does a lot of experimenting, which is why I gave it a original POSITIVE rating. However, the article's flaws are really obvious, and I hope that the author will thoroughly improve them based on the comments. I will further check the rebuttal.


**Strengths And Weaknesses:**

(a) Model
+ The use of "graphs" for multimodal pre-training is practical. In essence, the model benefits from having strong reasoning skills.
- However, the details of the model are not well presented, including data processing, sample visualization, etc.

(b) Experiments
+ The experiments in the article are adequate and verify that the learning graph can help the model improve multimodal learning.
- Which of these loss functions affects the results the most, and is it graph-based loss? If not, the model appears to be oriented east-west.
- In Figure 1, because the template "not a photo of ｛}" is inherently misleading. In addition, the car below is between yellow and green in color. As a result, this is not a particularly good example to demonstrate what the author is trying to say.
- There is not much evidence to demonstrate that the proposed methods can really help the model for modal alignment. It is difficult to verify this conclusion just by show only the SOTA results (without some newest SOTA baselines ) and lacking visual graphs/tables.
- Except for VQA, there is no obvious advantage of the effect of the model in this paper.

(c) writing
+ The reviewer likes the motivation in the Introduction section.
- There aren't enough ablation experiments to confirm that every design is rational.
- Many earlier works, such as VLPs in e-commerce, also address the issue of alignment from different techniques.  The "co-occurence" is the key factor that is the same with this article (no matter the word-patch alignment, or using RoI-tags). The paper's references, however, are insufficient.

(d) others
- The SOTA performance was obtained by ignoring a large number of SOTA models. The authors  should give a justifiable explanation.


Update after rebuttal

Thanks for your detailed response. I stick to my score.

---

> ### Author Response · Authors · 2022-08-02
> **Response to Reviewer WaXR (Part 5)**
>
> **6. The paper's references, are insufficient.** *Many earlier works, such as VLPs in e-commerce, also address the issue of alignment from different techniques. The "co-occurence" is the key factor that is the same with this article (no matter the word-patch alignment, or using RoI-tags).*
>
> Thanks for the valuable suggestion. We have updated the manuscript with a new subsection in the related works section regarding the concept alignment techniques in the multi-modal pretraining framework and provided a discussion on the connection with our work.
>
> The problem of semantic misunderstanding has also been investigated by previous works. EI-CLIP [1] considers the problem of cross-modal retrieval in the field of E-commerce. Sharing similar insight with our work, the authors notice the model bias towards some specific word tokens in CLIP and introduce causal inference to align the text encoder with e-commerce domain knowledge. K3M [2] focuses on the modality-missing and modality-noise problem and introduces knowledge modality into E-commerce tasks. DeVLBert [3] studies the spurious correlations between different modalities and adjusts the conditional probability of image tokens and word tokens. Kaleido-BERT [4] focuses on image-text coherence by introducing several novel self-supervised tasks.
>
> Comparably, the aforementioned works still relied on the simple image-text relation, and attempt to amend the relation to an unbiased form, while our approach forces the model to learn complicated relations, e.g., position relation, attributes, and dependencies, which would be more suitable under real-world scenarios.
>
> [1] Ma, Haoyu, et al. "EI-CLIP: Entity-Aware Interventional Contrastive Learning for E-Commerce Cross-Modal Retrieval." *Proceedings of the IEEE/CVF Conference on Computer Vision and Pattern Recognition*. 2022.
>
> [2] Zhu, Yushan, et al. "Knowledge perceived multi-modal pretraining in e-commerce." *Proceedings of the 29th ACM International Conference on Multimedia*. 2021.
>
> [3] Zhang, Shengyu, et al. "DeVLBert: Out-of-distribution Visio-Linguistic Pretraining with Causality." *Proceedings of the IEEE/CVF Conference on Computer Vision and Pattern Recognition*. 2021.
>
> [4] Zhuge, Mingchen, et al. "Kaleido-bert: Vision-language pre-training on fashion domain." *Proceedings of the IEEE/CVF Conference on Computer Vision and Pattern Recognition*. 2021.
>
>
>
> **7. The reviewer suggests the authors take the time to consider why Multimodal Pre-training requires Graph (or knowledge)**
>
> As we have stated in the introduction section, we believe that the simple image-text form of training samples leads current vision-language pre-training models to a trivial solution by relying on the co-occurrence of concepts. The lack of perception of semantic meanings may lead to inferior performances on several tasks. Unlike previous approaches, we augment the simple match/unmatch relation to a set of complicated relations, which finally proved to be effective when transferring to downstream tasks.
>
>
>
> **8. The authors claimed that focusing on the graph can help semantic alignment at first, but the second half of the paper, just shows that continue learning with graph will have some improvement to CLIP.**
>
> Sorry for the confusion. What we are trying to emphasize is the practical improvement brought by our work. In fact, we see our work as a flexible framework that can be applied to various pre-trained models, where we use CLIP as a showcase to demonstrate its effectiveness. Therefore, we mainly focus on showing the improvements with the vanilla CLIP which in turn reflect the potential gain for other pre-trained models.
>
> On the other hand, current large-scale pre-training approaches usually consume massive computation costs. Therefore, we consider adopting a continuous learning scheme as a simple implementation approach to achieve higher training efficiency. The added ablation study further proves that simply training CLIP with extra data and time fails to provide better results, showing that the improvements actually come from the injection of knowledge.
>
>
>
> Lastly, we are grateful for your time and we hope we have adequately addressed your concerns. If we have any other concerns, please let us know. We look forward to additional discussion.

---

> ### Author Response · Authors · 2022-08-02
> **Response to Reviewer WaXR (Part 4)**
>
> **5. Experiment**
>
> **5.1 The SOTA performance was obtained by ignoring a large number of SOTA models.** *The authors should give a justifiable explanation. (see more experiments)*
>
> The reviewer pointed out that some competitive baselines are not included in the experiment section. We argue that we have included most of the advanced approaches that were published before the submission of NeurIPS2022 (usually new published work near the submission deadline should not be considered, e.g., published in CVPR2022.) Nevertheless, we provide a comparison with additional baselines including, ViLT, Uni-Perceiver, and FLAVA and show the results on retrieval and VQA task below.
>
> Tab. 3 Fine-tuned image-text retrieval results on Flockr30K and COCO datasets. The better result between CLIP and our approach is shown in bold.
>
> |               |         Flickr30K          |          Flickr30K          |           MSCOCO           |           MSCOCO            |
> | ------------- | :------------------------: | :-------------------------: | :------------------------: | :-------------------------: |
> | Method        | Text Retrieval (R1/R5/R10) | Image Retrieval (R1/R5/R10) | Text Retrieval (R1/R5/R10) | Image Retrieval (R1/R5/R10) |
> | UNITER        |    87.3  / 98.0 /  99.2    |    75.6  / 94.1 /  96.8     |    65.7  / 88.6 / 93.8     |     52.9  / 79.9 / 88.0     |
> | VILLA         |   87.9  / 97.5  /  98.8    |    76.3 /  94.2  /  96.8    |             -              |              -              |
> | OSCAR         |             -              |              -              |     73.5 / 92.2 / 96.0     |    57.5 /  82.8 /  89.8     |
> | ERNIE-ViL     |   88.7  / 98.0  /  99.2    |    76.7 /  93.6  /  96.4    |             -              |              -              |
> | Unicoder-VL   |     86.2 / 96.3 / 99.0     |     71.5 / 91.2 / 95.2      |     62.3 / 87.1 / 92.8     |     48.4 / 76.7 / 85.9      |
> | ViLT          |     83.5 / 96.7 / 98.6     |     64.4 / 88.7 / 93.8      |     61.5 / 86.3 / 92.7     |     42.7 / 72.9 / 83.1      |
> | Uni-Perceiver |     87.9 / 98.2 / 99.1     |     74.9 / 93.5 / 96.0      |     64.7 / 87.8 / 93.7     |     48.3 / 75.9 / 84.5      |
> | CLIP          |    88.6 /  98.5 / 99.4     |     72.4 / 92.3 /  96.6     |    67.3  / 85.4 / 92.4     |     54.3 / 83.5 / 90.0      |
> | Ours          |   **89.2 / 98.9 / 99.4**   |   **75.7 /  94.4 / 96.8**   |   **70.2 / 89.2 / 94.4**   |   **57.6 / 83.9 / 90.4**    |
>
> Tab. 4 Fine-tuned image-text retrieval results on VQA dataset. The better result between CLIP and our approach is shown in bold.
>
> |               |         VQA         |
> | ------------- | :-----------------: |
> | Method        | test-dev / test-std |
> | UNITER        |    72.70 / 72.91    |
> | VILLA         |    73.59 / 73.67    |
> | OSCAR         |    73.16 / 73.44    |
> | ALBEF         |    74.54 / 74.70    |
> | Uni-Perceiver |     73.4 / 74.1     |
> | FLAVA         |      72.8 / -       |
> | CLIP          |    74.10 / 73.56    |
> | Ours          |  **76.11 / 75.24**  |
>
> We have updated the comparison in the main paper (see Tab.1 and Tab.2 in main paper).
>
> **5.2 Except for VQA, there is no obvious advantage of the effect of the model in this paper.**
>
> Despite that Knowledge-CLIP does not achieve SOTA performances under some settings, we believe the results still demonstrate the effectiveness and value of our work from the following perspectives:
>
> (1) Compared to the vanilla CLIP, our model achieves consistent improvements under **all** **settings**. Besides VQA, our model shows 2~3% higher performances on the retrieval task (R@1 metric), 1% higher on the entailment dataset SNLI-VE, and an average of 2.2% higher performance on 7 tasks on the GLUE dataset. This proves that injecting knowledge into multi-modal pre-training can practically improve generalization performance;
>
> (2) Our framework is not limited to injecting knowledge information into the CLIP models, as our training objectives and new knowledge graph datasets are technically compatible with other large-scale pretraining frameworks, e.g., Uni-Perceiver, ALBEF,... In this way, our work provides valuable direction on further improving the performances of other SOTA approaches.

---

> ### Author Response · Authors · 2022-08-02
> **Response to Reviewer WaXR (Part 3)**
>
> **4. There is not much evidence to demonstrate that the proposed methods can really help the model for modal alignment.** *It is difficult to verify this conclusion just by show only the SOTA results (without some newest SOTA baselines ) and lacking visual graphs/tables.*
>
> Thanks for the valuable suggestion. Specifically, we have updated several experiments that may help to understand the effectiveness of our framework.
>
> First, we show some visualization results on several downstream tasks, especially regarding the cases involving complicated semantic descriptions. The results in Figure 1-3 (supplementary) show that our method can improve the model's ability on understanding object attributes like color/scale and the relative position between objects. The results in Figure 4 (supplementary) show that our model can handle images with multiple objects, e.g., row 1 and 2, and handle actions/interactions between objects, e.g., row 3 and 4. The results in Figure 5 (supplementary) show that our model can understand the negation, e.g., rows 1 and 2, and has better reasoning ability.
>
> Second, we conduct experiments on carefully selected data that may better reflect how a vision-language model understands a particular type of input. Specifically, we select questions in the VQA dataset that contains (1) Negation including: 'not' / 'no' / 'nothing' / 'unlikely' (2) Color attribute including : 'green' / 'red' / 'yellow' / 'white' (3) Position attribute: 'between' / 'behind' / 'along' / 'holding' / 'in front of / 'sitting on' / 'standing on' / 'under' (4) Size: 'small' / 'smaller' / 'smallest' / 'big' / 'bigger' / 'biggest' / 'tiny' / 'huge' / 'medium'. We summarize the comparison results of CLIP and our model on these sub-datasets below.
>
> Table 2 Ablation studies on particular type of input.
>
> | **Datasets** | **CLIP** | **Knowledge-CLIP** |
> | :----------: | :------: | :----------------: |
> |   Negation   |   64.7   |    66.8 (+2.1)     |
> |    Color     |   54.2   |    59.9 (+5.7)     |
> |   Position   |   61.2   |    68.3 (+7.1)     |
> |     Size     |   72.1   |    73.4 (+1.3)     |
>
> As we can observe, our model achieves consistent improvements over CLIP on these specially designed datasets and shows significantly better results. Regarding questions with negation, our model achieves 2.1% higher accuracy. Regarding color and position attributes, our model shows even higher improvements. We believe these comparisons on different 'semantic domains' demonstrate the effectiveness of injecting knowledge information into the current vision-language pre-training framework which practically enhances the model perception on semantic understanding.
>
> We have updated the results and discussion in the supplementary (Sec.O.4).

---

> ### Author Response · Authors · 2022-08-02
> **Response to Reviewer WaXR (Part 2)**
>
> We show the comparison results on two representative tasks above, including the image/text retrieval task on Flickr30K, and the visual question answering task in VQA. Several observations can be made from the ablation:
>
> (1) By comparing the first and second row, we can see that simply training the CLIP model with extra time and data fails to improve the generalization performance. It also demonstrates that the improvements mainly come from the injected knowledge information rather than the continuous learning scheme.
>
> (2) All three training objectives (E2E, E2R, G2E) contribute to improving the model performance. Training the model without any of the objectives leads to inferior performances on downstream tasks. Specifically, we would like to clarify that all three losses are designed to use knowledge information. Despite the G2E loss that extracts structural features directly from the graph, E2E and E2R losses concentrate on using triplets derived from graphs, which are also proved to contain rich knowledge information in the field of knowledge reasoning. Therefore, we believe all three training objectives are in accord with our motivation and contribute to injecting knowledge into the vision-language model.
> We have updated the results and detailed discussion in the revised version of the supplementary material (Sec.O.1, Tab.1).
>
> **3. In Figure 1**, *because the template "not a photo of ｛}" is inherently misleading. In addition, the car below is between yellow and green in color. As a result, this **is not a particularly good example to demonstrate what the author is trying to say.***
>
> Sorry for the confusion. To better illustrate our claim, we give several additional toy examples in the supplementary material to show how the vanilla CLIP model handles semantic inputs.
>
> The first example (shown in Supplementary Figure 2) contains an image with two main objects: a **white** car and a **red** house. In this case, we consider two templates including 'a photo of a white {}' and 'a photo of a red {}'.  Vanilla CLIP still tends to provide similar outputs and recognize the same object in the image. This proves that vanilla CLIP fails to understand the meaning of color descriptions.
>
> The second example (shown in Supplementary Figure 3) considers scenarios with size and location descriptions. Given a photo of a strawberry and an apple, we use the template of 'a photo of small {} and big {}' and 'a photo of {} on the left and {} on the right' as the input. In this case, we constrain the candidate text token to {apple, strawberry} to better reflect the model bias. As a result, CLIP also fails to understand the semantic meaning and recognizes the relative position/scale of the objects.
>
> We also provide toy examples in the VQA task (shown in Supplementary Figure 5) to show how CLIP understands the questions with negation. As we can see from the first example in Figure 5, given the question 'What is the street number not name?', CLIP has higher confidence in predicting the name instead of the number. A similar situation happens in the second example, where CLIP tends to rely on the co-occurrence and predict dumping/dumping trash.
>
> We believe the aforementioned examples can help support our claim that the image-text training scheme in CLIP fails to provide semantic perceptions, and injecting knowledge information may be a feasible direction. We also provide the prediction of our method in these examples and show that a knowledge-based training scheme can practically help model perception on these semantic descriptions. If the reviewer finds the newly provided examples are more suitable to illustrate our motivation, we will update them in Figure 1 (main paper).

---

> ### Author Response · Authors · 2022-08-02
> **Response to Reviewer WaXR (Part 1)**
>
> We would first like to express our appreciation for your time and insightful comments. Please find our response to your concerns in the following:
>
> **1. The details of the model are not well presented** *including data processing, sample visualization, etc.*
>
> Thanks for the valuable suggestion.
>
> For data processing, the five datasets we used are all public datasets that have been widely used in early works. Therefore, we practically follow the data processing routine. Specifically, for VisualSem, each concept (entity) in the triplet has both corresponding images and text descriptions and will be randomly chosen if the triplet is sampled. In this way, the modality of the concept in different triplets or training batches can be different, and the triplet forms can include {image/text, relation, image/text}. Differently, the Visual Genome dataset contains scene graphs for each image. The nodes are presented in a bounding box and the edges are represented by word tokens, e.g., standing on. We extract the image features of the corresponding box and generate {image, relation, image} triplets. For each image in Visual Genome, we randomly sample 4 triplets, based on the consideration that a larger number may lead to repeated sampling. The triplets in ConceptNet are pre-processed and explicitly given by the authors. So we directly sample them in the training batch. For CC3M and COCO Caption, we convert the original image-text pairs to triplets by adding self-defined semantic relations ’image of’ and ’caption of'.
>
> For each input modality in the training data, we adopt a unified processing procedure to make it possible for batch training. Specifically, the length of the image is set as 16x16 and the length of the text is set as 77. We adopt the same data augmentation as vanilla CLIP including resize, center crop, and normalization for images. For text, a start of text token and an end of text token are first concatenated with the input and the BPE tokenizer is adopted to encode the words. For each training batch, 75% of data is sampled from the three knowledge graph datasets, and 25% of data is sampled from CC3M/COCO Caption.
>
> We have updated the detailed configurations in the supplementary (Sec.O.5).
>
> The visualization of some training data is shown in Figure 3 (main paper). To better illustrate, we provide additional examples in supplementary material and show some visualization results on downstream tasks. Please refer to Figure 1-5 in the supplementary.
>
>
>
> **2. Which of these loss functions affects the results the most** *and is it graph-based loss? If not, the model appears to be oriented east-west.* T*here aren't enough ablation experiments to confirm that every design is rational.*
>
> To validate the effectiveness of the components in our work, we carefully design several settings, including:
>
> (1) CLIP+continuous learning: we train vanilla CLIP (pretrained weights as initialization) on knowledge datasets adopted in our work.
>
> (2) Knowledge-CLIP-(t1, t2, t3): we remove the training objectives respectively in our work to analyze the contribution of each loss.
>
> Considering the time limitation, we adopt a smaller model (ViT-B/32) as the image encoder of CLIP in the ablation study and will update the full model ablation in the future. Also, it is worth noticing that KD loss (see sec.4.3 in the main paper) plays a vital role in the continuous learning scheme, without which will lead to a significant performance drop due to the model forgetting problem. Therefore, we use KD loss in all the ablation settings for a fair comparison.
>
> Table 1 Ablation studies of continuous learning/training objectives. We report results on the Flickr30K and VQA.
>
> | **Model**                | **KG** **datasets** | **E2E Loss** | **E2R Loss** | **G2E Loss** | **Flickr30K** Retrieval |       **VQA**       |
> | ------------------------ | :-----------------: | :----------: | :----------: | :----------: | :---------------------: | :-----------------: |
> |                          |                     |              |              |              |      Text / Image       | test-dev / test-std |
> | CLIP                     |          -          |      -       |      -       |      -       |       84.2 / 63.1       |     68.9 / 69.2     |
> | CLIP+Continuous Learning |        **√**        |      -       |      -       |      -       |       84.5 / 63.0       |     69.1 / 69.5     |
> | Knowledge-CLIP-t1        |        **√**        |      -       |    **√**     |    **√**     |       85.0 / 64.6       |     70.4 / 71.1     |
> | Knowledge-CLIP-t2        |        **√**        |    **√**     |      -       |    **√**     |       85.7 / 66.0       |     71.2 / 69.9     |
> | Knowledge-CLIP-t3        |        **√**        |    **√**     |    **√**     |      -       |       84.9 / 65.8       |     70.2 / 70.4     |
> | Knowledge-CLIP (Full)    |        **√**        |    **√**     |    **√**     |    **√**     |     **86.3 / 67.2**     |   **72.5 / 72.7**   |

---

### Official Review · Reviewer_zySD · 2022-07-12

**Rating:** 8
**Confidence:** 4
**Soundness:** 3 good
**Presentation:** 3 good
**Contribution:** 3 good

**Summary:**

This paper presents a vision-language pre-training framework that incorporates knowledge information by pre-training on multiple knowledge graph datasets (i.e., VisualSem, Visual Genome, and ConceptNet).
They unify all the possible triples of different modalities (e.g., image-relation-text, image-relation-image, text-relation-text), encode each modality with its corresponding CLIP encoder (i.e., Image encoder, Text encoder), and concatenate all the embeddings with special token <head>.
Then, multi-modal encoder (i.e., 4 layer transformer model) encodes it and uses embedding of <head> token as a representation.
For the training objectives, the paper utilizes three different objectives including entity-entity, entity-relation, and graph-entity loss.
The pre-trained model consistently outperforms baselines including UNITER, OSCAR, CLIP, ALBEF on various multi-modal tasks.

**Questions:**

1. Figure 2: L_E2L -> L_E2R ?

**Ethics Review Area:**

["I don’t know"]

**Limitations:**

I really like the approach but the lack of analysis is the limitation of this paper.

I am willing to increase the score when more analysis about training objectives are provided.

**Strengths And Weaknesses:**

### Strengths

1. It is interesting that how framework encodes multiple features of different modalities into one representation.
2. The performance shows the effectiveness of the approach.

### Weakness

1. Lack of analysis
    - Want to see the performance with each training objective for understanding the effectiveness of each objective. (e.g., without L_E2E, without L_E2R, without L_G2E)
    - Want to see the impact of L_KD, which is the loss term of KL distance between original CLIP and the proposed model.

2. Lack of intuition for each objective
    - E2E: Want to see the intuition behind exploiting contrastive learning between (head, relation)-masked triple and (tail)-masked triple.
    - G2E: Want to see the intuition behind applying contrastive learning between before and after graph-propagation on (tail).

---

> ### Author Response · Authors · 2022-08-02
> **Response to Reviewer zySD (Part 2)**
>
> **2. Lack of intuition for each objective.** *E2E: Want to see the intuition behind exploiting contrastive learning between (head, relation)-masked triple and (tail)-masked triple; G2E: Want to see the intuition behind applying contrastive learning between before and after graph-propagation on (tail).*
>
> We figure that E2E loss practically emphasizes semantic understanding of the concepts. For example, given a triplet {teacher, teaches, student}, the phrase 'teacher teaches' and the word 'students' in fact have similar semantic meanings. In this case, we use the contrastive loss to close the gap between the feature representations of semantic-similar concepts. The entity prediction task in the field of knowledge graph shares a similar insight aiming to predict the missing tail entity in one triple (h, r, ?) -> (h, r, t), which is viewed as an essential prerequisite for reasoning.
>
> G2E loss, on the other hand, exploits the structural information in the knowledge graph dataset. The contrastive learning objective can in fact be viewed as a node classification task where the output feature from graph layers (after extracting structural information) serves as the ground truth label for each node. In this way, we force the model to learn structure-injected features, which in turn promotes the reasoning process through the whole graph.
>
> Generally, we believe the E2E, E2R, and G2E loss promote the model from different perspectives by focusing on **semantic understanding of concepts**, **complicated relations between entities**, and **structural information** respectively. The ablation studies on all the training objectives also prove that each loss gives a certain contribution.
>
>
>
> **3. Figure 2: L_E2L -> L_E2R ?**
>
> Thanks for pointing out. We have updated it in the revised version.
>
>
>
> Lastly, we are grateful for your time and we hope we have adequately addressed your concerns. If we have any other concerns, please let us know. We look forward to additional discussion.

---

> > ### Comment · Reviewer_zySD · 2022-08-10
> > **Response to Author**
> >
> > Thanks to the authors for answering questions and providing requested analysis.
> > Motivation and impact of each objective are much clear now.
> > I have raised my rating. Thanks for your contribution.

---

> ### Author Response · Authors · 2022-08-02
> **Response to Reviewer zySD (Part 1)**
>
> We would first like to express our appreciation for your time and insightful comments. Please find our response to your concerns in the following:
>
> **1. Lack of analysis.** *Want to see the performance with each training objective for understanding the effectiveness of each objective. (e.g., without L_E2E, without L_E2R, without L_G2E); Want to see the impact of L_KD, which is the loss term of KL distance between original CLIP and the proposed model.*
>
> Thanks for the valuable suggestion. To validate the effectiveness of the components in our work, we carefully design several settings, including:
>
> (1) Knowledge-CLIP-(t1, t2, t3): we remove the training objectives respectively in our work to analyze the contribution of each loss.
>
> (2) Knowledge-Knowledge-CLIP w/o KD: we remove the KD loss in the training objective and keep the other three losses unchanged.
>
> Considering the time limitation, we adopt a smaller model (ViT-B/32) as the image encoder of CLIP in the ablation study and will update the full model ablation in the future.
>
> Table 1 Ablation studies of training objectives. We report results on the Flickr30K retrieval task and VQA task with ViT-B/32 as image encoder.
> |                       |                 |         |         |         |        | **Flickr30K** Retrieval |       **VQA**       |
> | --------------------- | :-------------: | :-----: | :-----: | :-----: | :----: | :---------------------: | :-----------------: |
> | **Model**             | **KG** **data** | **E2E** | **E2R** | **G2E** | **KD** |      Text / Image       | test-dev / test-std |
> | Knowledge-CLIP-t1     |      **√**      |    -    |  **√**  |  **√**  | **√**  |       85.0 / 64.6       |     70.4 / 71.1     |
> | Knowledge-CLIP-t2     |      **√**      |  **√**  |    -    |  **√**  | **√**  |       85.7 / 66.0       |     71.2 / 69.9     |
> | Knowledge-CLIP-t3     |      **√**      |  **√**  |  **√**  |    -    | **√**  |       84.9 / 65.8       |     70.2 / 70.4     |
> | Knowledge-CLIP w/o KD |        √        |    √    |    √    |    √    |   -    |       82.4 / 62.5       |     67.4 / 66.6     |
> | Knowledge-CLIP (Full) |      **√**      |  **√**  |  **√**  |  **√**  | **√**  |     **86.3 / 67.2**     |   **72.5 / 72.7**   |
>
> We show the comparison results on two representative tasks above, including the image/text retrieval task on Flickr30K, and the visual question answering task in VQA. Several observations can be made from the ablation:
>
> (1) All three training objectives (E2E, E2R, G2E) contribute to improving the model performance. Training the model without any single objective leads to inferior performances on downstream tasks. We argue that the E2E, E2R, and G2E loss promote the model from different perspectives by focusing on **semantic understanding of concepts**, **complicated relations between entities**, and **structural information.** Therefore, all three objectives are necessary for the framework and contribute to the improvement respectively.
>
> (2) The model achieves lower after removing the KD loss, indicating its vital role in the continuous learning scheme. We argue the reason for this phenomenon is that the model suffers from the forgetting problem, which is widely spotted in the field of lifelong learning / continuous learning.
>
> We have updated the results and detailed discussion in the revised version of the supplementary material (Sec.O.1).

---

### Official Review · Reviewer_7sfH · 2022-07-25

**Rating:** 5
**Confidence:** 3
**Soundness:** 2 fair
**Presentation:** 3 good
**Contribution:** 2 fair

**Summary:**

The paper extends the Contrastive Language-Image Pre-training (CLIP) model with a knowledge-based pre-training framework. The framework introduces knowledge-based objectives in the pre-training process and utilizes diverse types of knowledge graphs as training data.

Contributions:
- A framework to utilize knowledge graph to vision-language pretraining
- Continued pretraining to save computation resources
- Improved performance upon CLIP on vision-language multi-modal tasks as well as uni-modal tasks.

**Questions:**

- Equation 13 is confusing to me. Does it have a simple but meaningless solution that G^L(t) = Y(-,-,t)?


**Limitations:**

The authors adequately addressed the limitations and potential negative societal impact of their work.

**Strengths And Weaknesses:**

Strengths:

- The framework is intuitive and clearly presented.
- The improvement of the CLIP model is significant and promising.

Weakness:
- The lack of proper ablation study. For the vision-language or uni-modal tasks in the evaluation, not much reasoning is required to perform the task. So, it is important to answer the question: if the improvement in Knowledge-CLIP comes from the knowledge-based objective or just from more training data. An ablated baseline would be CLIP continued pretrained on the same datasets which Knowledge-CLIP was pretrained.
- Another ablation study should be on the three objectives to measure their contributions to the final result.
- The motivation in figure 1 is not accurate. The example only shows that CLIP fails to capture negation, not all semantics. From the same figure, we can see that CLIP performs well in differentiating cars from planes. Authors need to be more specific on their claims.

---

> ### Author Response · Authors · 2022-08-02
> **Response to Reviewer 7sfH (Part 2)**
>
> **2. The motivation in figure 1 is not accurate.** *The example only shows that CLIP fails to capture negation, not all semantics. From the same figure, we can see that CLIP performs well in differentiating cars from planes. Authors need to be more specific on their claims.*
>
> Sorry for the confusion. We fully agree that CLIP does well at recognizing objects, e.g., differentiating cars from planes. Nevertheless, what we are trying to convey from Figure 1 is that CLIP also has limitations from other aspects. For example, when dealing with more fine-grained descriptions, e.g., logical descriptions like negation or object attributes like colors, the CLIP model usually fails to understand the semantic meanings of them and still relies on the 'object' itself. As we have illustrated in Figure 1 in the main paper, the descriptions 'a photo of airliner' and 'not a photo of airliner' both have a strong correlation with the airliner image, the descriptions of yellow and green are also similarly correlated to the image of yellow sports car. This indicates that the current model struggles with **semantic** language input, which leads to inferior performances on tasks that require reasoning ability. These observations motivate us to inject knowledge into the current training scheme and address the limitations.  We have clarified our claim in the Introduction and caption of Figure 1 (in the revised version).
>
> To better illustrate our claim, we give two additional toy examples in the supplementary material to show how the vanilla CLIP model handles semantic inputs. The first example (shown in Supplementary Figure 2) contains an image with two main objects: a **white** car and a **red** house. In this case, we consider two templates including 'a photo of a white {}' and 'a photo of a red {}'. Vanilla CLIP still tends to provide similar outputs and recognize the same object in the image. This proves that vanilla CLIP fails to understand the meaning of color descriptions.
>
> The second example (shown in Supplementary Figure 3) considers scenarios with size and location descriptions. Given a photo of a strawberry and an apple, we use template of 'a photo of **small** {} and **big** {}' and 'a photo of {} on the **left** and {} on the **right**' as the input. In this case, we constrain the candidate text token to {apple, strawberry} to better reflect the model bias. As a result, CLIP also fails to understand the semantic meaning and recognizes the relative position/scale of the objects.
>
> We believe the aforementioned examples can help support our claim that the image-text training scheme in CLIP fails to provide semantic perceptions, and injecting knowledge information may be a feasible direction. We also provide the prediction of our method in these examples and show that a knowledge-based training scheme can practically help model perception on these semantic descriptions.
>
>
>
> **3. Equation 13 is confusing to me.** *Does it have a simple but meaningless solution that G^L(t) = Y(-,-,t)?*
>
> We believe the model would not collapse to this trivial solution from two aspects.
>
> (1) In fact, the $G^L(t)$ term in Eq.(13) is derived from Eq.(11-12) where $G^L(t)$ aggregates the features from a local neighborhood. Therefore, $G^L(t)$ will be presented as the function of a set of features $\{Y(\text{-},\text{-},t_i)\}$ rather than $Y(\text{-},\text{-},t)$ alone.
>
> (2) The contrastive loss G2E drives the model simultaneously maximizing the similarity of positive pair, i.e., $cos(G^L(t_i), Y(\text{-},\text{-},t_i))$, and minimizing the similarity of negative pair, i.e., $cos(G^L(t_j), Y(\text{-},\text{-},t_i))$. The mentioned trivial solution only meets the demand of the first half, and may not be an optimal solution for the whole objective.
>
> Lastly, we are grateful for your time and we hope we have adequately addressed your concerns. If you have any other concerns, please let us know. We look forward to additional discussion.

---

> ### Author Response · Authors · 2022-08-02
> **Response to Reviewer 7sfH (Part 1)**
>
> We would first like to express our appreciation for your time and insightful comments. Please find our response to your concerns in the following:
>
> **1. The lack of proper ablation study.** *For the vision-language or uni-modal tasks in the evaluation, not much reasoning is required to perform the task. So, it is important to answer the question: if the improvement in Knowledge-CLIP comes from the knowledge-based objective or just from more training data. An ablated baseline would be CLIP continued pretrained on the same datasets which Knowledge-CLIP was pretrained. Another ablation study should be on the three objectives to measure their contributions to the final result.*
>
> Thanks for the valuable suggestion. To validate the effectiveness of the components in our work, we carefully design several settings, including:
>
> (1) CLIP+continuous learning: we train vanilla CLIP (pretrained weights as initialization) on knowledge datasets adopted in our work.
>
> (2) Knowledge-CLIP-(t1, t2, t3): we remove the training objectives respectively in our work to analyze the contribution of each loss.
>
> Considering the time limitation, we adopt a smaller model (ViT-B/32) as the image encoder of CLIP in the ablation study and will update the full model ablation in the future. Also, it is worth noticing that KD loss (see sec.4.3 in the main paper) plays a vital role in the continuous learning scheme, without which will lead to a significant performance drop due to the model forgetting problem. Therefore, we use KD loss in all the ablation settings for a fair comparison.
>
> Table 1 Ablation studies of continuous learning / training objectives. We report results on the Flickr30K retrieval task and VQA task with ViT-B/32 as image encoder.
> | **Model**                | **KG** **datasets** | **E2E Loss** | **E2R Loss** | **G2E Loss** | **Flickr30K** Retrieval |       **VQA**       |
> | ------------------------ | :-----------------: | :----------: | :----------: | :----------: | :---------------------: | :-----------------: |
> |                          |                     |              |              |              |      Text / Image       | test-dev / test-std |
> | CLIP                     |          -          |      -       |      -       |      -       |       84.2 / 63.1       |     68.9 / 69.2     |
> | CLIP+Continuous Learning |        **√**        |      -       |      -       |      -       |       84.5 / 63.0       |     69.1 / 69.5     |
> | Knowledge-CLIP-t1        |        **√**        |      -       |    **√**     |    **√**     |       85.0 / 64.6       |     70.4 / 71.1     |
> | Knowledge-CLIP-t2        |        **√**        |    **√**     |      -       |    **√**     |       85.7 / 66.0       |     71.2 / 69.9     |
> | Knowledge-CLIP-t3        |        **√**        |    **√**     |    **√**     |      -       |       84.9 / 65.8       |     70.2 / 70.4     |
> | Knowledge-CLIP (Full)    |        **√**        |    **√**     |    **√**     |    **√**     |     **86.3 / 67.2**     |   **72.5 / 72.7**   |
>
> We show the comparison results on two representative tasks above, including the image/text retrieval task on Flickr30K, and the visual question answering task in VQA. Several observations can be made from the ablation:
>
> (1) All three training objectives (E2E, E2R, G2E) contribute to improving the model performance. Training the model without any single objective leads to inferior performances on downstream tasks. We argue that the E2E, E2R, and G2E loss promote the model from different perspectives by focusing on **semantic understanding of concepts**, **complicated relations between entities**, and **structural information.** Therefore, all three objectives are necessary for the framework and contribute to the improvement respectively.
>
> (2) By comparing the first and second row, we can see that simply training the CLIP model with extra time and data fails to improve the generalization performance. It also demonstrates that the improvements mainly come from the injected knowledge information rather than the continuous learning scheme.
>
> We have updated the results and detailed discussion in the revised version of the supplementary material (Sec.O.1, Tab.1).

---

### Author Response · Authors · 2022-08-02
**General Response to All Reviewers**

We thank all the reviewers for their insightful and valuable comments.

We have revised the manuscript according to the reviewers' comments. The main changes we made include:

1. Supplementary Section O.1: we include ablation studies on the training objectives and the continuous learning scheme.
2. Supplementary Section O.2: we include additional toy examples to show the comparison between CLIP and our method.
3. Supplementary Section O.3: we include visualization results on downstream tasks.
4. Supplementary Section O.4: we include experiments to analyze the performance of our model on particular semantics.
5. Supplementary Section O.5: we include some implementation details.

All the responses in text and table are presented below. Nevertheless, due to the reason that images cannot be uploaded to the openreview system, we apologize for the inconvenience and hope the reviewers can refer to the revised supplementary material for the visualization results. All added images are shown in Section. O.

Lastly, we would like to thank the reviewers for their time and we are welcome for any further discussion.

---

### Meta-Review · Area_Chair_mEyQ · 2022-08-26

**Recommendation:** Accept
**Confidence:** Certain

**Metareview:**

Authors have improved their paper in such a way that multiple reviewers raised their original score. With varying degrees of excitement, the reviewers all agree that this paper should be accepted

Strengths:
- Interesting framework for multimodal pretraining integrating language, vision and knowledge graphs
- Strong empirical results on multiple benchmarks
- Ablation study demonstrates the benefit of each proposed component and objective
- Paper is well-written

Weaknesses:
- No consistent ones seem to remain, as the reviewers effectively responded to most of the concerns of reviewers in the rebuttal

**Award:**

No

---

### Decision · Program_Chairs · 2022-09-14

Accept